# Parental considerations about their childs' mental health: Validating the German adaptation of the Parental Reflective Functioning Questionnaire

**Andreas S. Wildner**[1]*, **Su Mevsim Küçükakyüz**[1], **Anton K. G. Marx**[1], **Tobias Nolte**[2], **Corinna Reck**[1], **Peter Fonagy**[2], **Patrick Luyten**[2], **Alexandra von Tettenborn**[1], **Mitho Müller**[1], **Anna-Lena Zietlow**[3], **Christian F. J. Woll-Weber**[1,4]

1 Clinical Psychology of Childhood and Adolescence & Counseling Psychology, Ludwig-Maximilians-Universität, Munich, Germany, 2 Division of Psychology and Language Sciences, Clinical, Education & Health Psychology, Psychoanalysis Unit, University College London, London, United Kingdom, 3 Institute of Clinical Psychology and Psychotherapy, Clinical Child and Adolescent Psychology, Technische Universität Dresden, Dresden, Germany, 4 Clinical Child and Adolescent Psychology and Psychotherapy, Freie Universität Berlin, Berlin, Germany

* andreas.wildner@psy.lmu.de

**Data Availability Statement:** The hypotheses, analysis plan, and cut-off criteria used in this study have been preregistered and are available along

## Abstract

### Introduction

Parental Reflective Functioning describes the parents' ability to view their child as motivated by mental states. The Parental Reflective Functioning Questionnaire (PRFQ) represents an 18-item and three-factor self-report measure. Our goal was to conduct the first German validation study.

### Method

In a community sample of 378 mothers of children aged 10.2–78.6 months, we used Confirmatory Factor Analysis with a cross-validation approach to assess model fit. Reliability was measured using Cronbach's $\alpha$ and McDonald's $\omega$. Concurrent validity was assessed using correlations with relevant constructs.

### Results

The three-factor structure of the original validation could be confirmed. The German model only needed minor modifications: two items had to be removed, and one error covariance was added. The resulting 16-item questionnaire with the three subscales "Pre-mentalizing", "Interest and Curiosity about Mental States", and "Certainty about Mental States" was successfully cross-validated (CFI = .94, TLI = .93, SRMR = .07, RMSEA = .04 (CI [.01, .06])). These factors were related in theoretically expected ways to parental attachment dimensions, emotional availability, parenting stress, and infant attachment status.

with the figures, analysis script, and all raw data necessary to reproduce the reported findings, under https://osf.io/j69wx/.

**Funding:** The author(s) received no specific funding for this work.

**Competing interests:** The authors have declared that no competing interests exist.

## Conclusion

While reliability could still be improved, the German 16-item version of the PRFQ represents a valid measure of parental reflective functioning.

## Introduction

### Parental reflective functioning and its relevance

Parental Reflective Functioning (PRF) relates to the parents' capacity to see their child as an intentional agent motivated by mental states, as well as the capacity to recognize how their own mental states affect their child [1]. As such, PRF was described as an application of mentalizing, namely the capacity to consider both others' (other-mentalizing) and one's own intentional mental states (self-mentalizing) [2]. PRF, as measured by the Parental Reflective Functioning Questionnaire (PRFQ) [3], is comprised of three dimensions. First, the Certainty about Mental States (CMS) captures the parents' awareness that mentalizing is inherently inferential and comes with some uncertainty. Second, the Interest and Curiosity about Mental States (IC) represents an attitude associated with more tolerance towards infants' distress [4,5] and better regulation of the parents´ own mental states [3]. Third, Pre-mentalizing (PM) aims to identify non-mentalizing modes, which would inhibit parents from entering the subjective world of their children [4] and can lead to misattributions of their children's actions [6].

PRF has been implicated in other relevant constructs such as children's development of Theory of Mind [7], epistemic trust [8], and attachment security [for an overview see 9]. For instance, Meins et al. [10] found PRF to predict child attachment in the Strange Situation Test. Overall, high PRF seems to increase the likelihood of secure attachment in children [11] and can act as a protective factor for development in children with early adversities [12]. On the mother's side, low PRF increases occurrences of maladapted maternal behavior [13]. This connection between PRF and attachment has been dubbed "loose coupling", describing the idea that secure attachment and emotional availability are not always, but often [14], associated with high PRF [15]. In practice, indicators of functional PRF like IC and CMS are less often related to secure attachment than indicators of dysfunctional PRF, such as PM [3], which represents one of the main reasons to include the PM dimension in the PRFQ. One explanation for this well-documented connection between PRF and attachment security might be how PRF changes the way parents interact with their children. PRF leads to more parental sensitivity towards the child and also towards distress displayed by the child [16,17]. This, in turn, is an important predictor of the emergence of secure attachment [18].

### The measurement of PRF

Historically, the majority of available measures for PRF and Mentalizing were interviews [19], such as a specialized version of the Adult Attachment Interview (AAI) or the Parent Development Interview (PDI). While these measures allow for very detailed information and data analysis, their time-intensive nature makes them unsuitable for large-scale assessments [3]. As such, the PRFQ is one of the first economic self-report scales of PRF [3]. The PRFQ, as developed and validated by Luyten et al. [3], is an 18-item self-report scale. It consists of the three subscales CMS, IC, and PM each containing six items measured on a 7-point Likert scale.

While there has been some uncertainty regarding the factor structure of the PRFQ, Luyten et al. [3] proposed that their 18-item questionnaire comprises three factors, of which only

CMS and IC are correlated. In their second study [3], this correlation was no longer significant. The model also contains several unspecified error covariances. To date, the PRFQ has been validated in various countries, including Portugal, Canada, South Korea, China, Italy, Hungary, Finland, and Iran [20–28]. Many of these studies found deviating factor structures, with factor counts ranging from two to five, and varying factor correlations, and as many as six items being removed. A more detailed discussion of these differences can be found in the online supplementary material.

## The current study

The present study has two goals: First, we aim to replicate the factor model of the original PRFQ. Second, we aim to validate the German version of the PRFQ. This German translation has been used first by Krink et al. [17]. After translating the PRFQ into German backtranslation was conducted by an English native speaker to ensure face validity. However, Krink et al. [17] only assessed the reliability as measured by Cronbach's $\alpha$ and, until today, this German adaptation has not been validated further [29]. While some publications have used the validation of the original PRFQ to demonstrate sufficient construct validity for the German translation [e.g., 30], this is inadvisable since it needs to be assured that a translation is not only verbally correct but also transmits the intended meaning, tone, and cultural differences [31]. To close this research gap, we aim to provide a full validation of the German translation of the PRFQ.

## Hypotheses

The following hypotheses have been formulated based on previous research by Campbell et al. [32] and Luyten et al. [3].

### Factor structure

<u>H1</u>: We assume a replication of the model of Luyten et al. [3]–Study 1. The original model provides a good model fit (H1.1) and has the same significant factors and loadings as Luyten et al. [3] (H1.2).

### Reliability

<u>H2</u>: All scales and subscales prove a reliable measure as indicated by both a Cronbach's $\alpha$ and a McDonald's $\omega$ value of $> .70$.

### Concurrent validity

To test the concurrent validity, a total of 30 individual hypotheses were preregistered, which can be found in the online supplement (for an overview see Supplementary Table 1 in S1 Table).

<u>H3</u>: We expect CMS, IC, and PM to be significantly correlated to relationship satisfaction.

We expect CMS to be correlated with the Epistemic Trust Mistrust and Credulity Questionnaire (ETMCQ). We expect PM to be significantly correlated to the ETMCQ, perceived stress,

postpartum depressiveness, education, and changes in the working environment due to COVID-19.

# Method

## Ethics statement

The study had been approved by the independent ethics committee of the medical faculty, Ruprecht-Karls-Universität, Heidelberg, in agreement with the ethics committee of the Ludwig-Maximilians-Universität, Munich (vote: S-446/2017) and the declaration of Helsinki 2013, seventh revision.

## Participants & procedure

The sample for this study is part of a longitudinal project called CoviFam, as part of a larger longitudinal, observational project called COMPARE [33]. In the CoviFam project, we investigate the well-being in a community sample of young families in the context of the COVID-19 pandemic. The initial recruiting for the CoviFam project took place from 06/05/2020 to 20/11/2020.

All data used in this study [34] stems from the third measuring time point, since previous timepoints did not contain the PRFQ. The used data was collected from the 23/02/2022 to the 06/05/2022. Participants were informed about the content of the study as well as their right to withdraw at any time. Afterwards, they gave their written, informed consent prior to completing an online questionnaire at a time and place of their choice.

The target population were mothers of children aged 0 to 3 years at the beginning of the longitudinal study, which were recruited for the study by various means such as social media channels (i.e., X/Twitter, Facebook and Instagram posts), medical institutions and professionals, and professional support organizations. We estimate that 53% of all participants have been made aware of the study via social media and the remaining 47% by word of mouth or flyers at health care facilities. Recruitment took place in Germany, Austria, and Switzerland. Next to country of residence and children's age the only other inclusion criterion was sufficient German language skills. The final size of this community sample was $N = 378$, after 11 cases had been removed due to missing data on the "Partnerschaftsfragebogen kurz" (Couples questionnaire short) or IDs entered erroneously by the mothers, following the specification of listwise deletion in our pre-registration. Another 40 participants were removed for indicating a gender other than female, to stick with the parameters used by Luyten et al. [3]. Thus, our sample missingness rate was 11 / (378 + 11 + 40) = 2.6%.

The data was collected before this validation was conceived, therefore followed no relevant stopping rules. However, we ensured that the sample size was sufficient by adhering to the convention of 10 participants per item [35] for both the training and validation dataset (10 x 18 x 2 = 360 < 378). The mothers' mean age was $M = 35.79$ ($SD = 4.37$, range = 19–50 years). The children's mean age was $M = 43.44$ months ($SD = 13.32$, range = 10.2–78.6). We included indicators of child wellbeing in the form of problems with crying, sleeping, and eating alongside the other sociodemographics in Table 1.

## Materials

The study consisted of an extensive collection of online-based self-report questionnaires. Means and standard deviations for all measures can be found in Supplementary Table 8 in S3 Table. Additionally, the following additional demographic variables were obtained: Children's and parents' age, the level of secondary education achieved, and if or if not the parents'

**Table 1. Participants' sociodemographic characteristics.**

| | n | % |
|---|---|---|
| Child's gender | 378 | |
| Female | 188 | 49.74 |
| Number of children in the family | 378 | |
| 1 child | 138 | 37.51 |
| 2 children | 182 | 48.15 |
| 3 children | 47 | 12.43 |
| 4 children | 9 | 2.38 |
| 5 children | 2 | 0.53 |
| Crying | 378 | |
| 1 (none) | 64 | 16.93 |
| 2 | 89 | 23.54 |
| 3 | 99 | 26.19 |
| 4 | 97 | 25.66 |
| 5 (often) | 40 | 10.58 |
| Issues with Sleeping (Child) | 378 | |
| 1 (none) | 140 | 37.04 |
| 2 | 107 | 28.31 |
| 3 | 65 | 17.20 |
| 4 | 48 | 12.70 |
| 5 (often) | 29 | 7.67 |
| Issues with Eating (Child) | 378 | |
| 1 (none) | 218 | 57.67 |
| 2 | 86 | 22.75 |
| 3 | 49 | 12.96 |
| 4 | 29 | 7.67 |
| 5 (often) | 7 | 1.85 |
| Relationship status | 378 | |
| Married (living together) | 319 | 84.39 |
| Relationship (living together) | 55 | 14.55 |
| Relationship (not living together) | 4 | 1.06 |
| School degree | 378 | |
| German middle school diploma* | 5 | 1.32 |
| German Realschule diploma | 38 | 10.05 |
| German Fachabitur | 49 | 12.96 |
| German Abitur | 286 | 75.66 |
| Monthly net income | 378 | |
| 0–1000 euro | 3 | 0.79 |
| 1000–2000 euro | 9 | 2.38 |
| 2000–3000 euro | 55 | 14.55 |
| 3000–5000 euro | 183 | 48.41 |
| > 5000 euro | 128 | 33.86 |
| Change in job situation due to the pandemic | 378 | |
| Working environment changed | 146 | 38.62 |

n = sample size.

*: German middle school = "Mittelschule" (Grade 5–9).

working environment had changed due to the COVID-19 pandemic. Most of our data stems from items measured on Likert scales, which are technically ordinally scaled. However, these ordinal scales may still be used for parametric tests [36] with little difference to nonparametric testing [37–40]. As such, we treated all Likert-scale questionnaires as interval scaled.

**Parental Reflective Functioning Questionnaire.** Parental Reflective Functioning was measured via the PRFQ by Luyten et al. [3]. The original PRFQ is an 18-item 3-factor self-report questionnaire using a 7-point Likert scale. The items of each subscale are averaged, yielding three mean values for each participant. The items used in the validation by Luyten et al. [3] as well as their translations for the present study can be found in Supplementary Tables 2–4 in S2 Table. In the original version, the reliability of PM was α = .70, for CMS α = .82, and for IC α = .75.

**Epistemic trust, mistrust, and credulity questionnaire.** The Epistemic Trust, Mistrust, and Credulity Questionnaire (ETMCQ) is a 18 item, three factor self-report questionnaire measuring epistemic trust on a 7-point Likert scale on three separate subscales for each participant [32]. As of yet, no validation of the German translation has been published. Reliability of the English version was α = .76–.81 for Trust, α = .65–.72 for Mistrust, and 75–.81 for Credulity. All items can be found in the Supplementary Table 5 in S2 Table.

**Perceived stress scale.** The German version of the Perceived Stress Scale [41] was used to measure stress. It is a 10 item, one-factor self-report questionnaire used on a 5-point Likert scale. After reverse coding, all items are averaged. Reliability has been reported at ω = .89. All items can be found in Supplementary Table 6 in S2 Table.

**Edinburgh postnatal depression scale.** The German version [42] of the Edinburgh Postnatal Depression Scale (EPDS) [43] was employed to measure depressiveness. While different factor structures have been used in the past, we opted for the 10-item, one-factor solution, measured on a 4-point Likert Scale. Values were simply reverse coded where necessary and then averaged. Reliability is α = .87. All items can be found in Supplementary Table 7 in S2 Table.

**Partnerschaftsfragebogen Kurz.** Lastly, the Partnerschaftsfragebogen Kurz (PFB-K) [44] was used to measure couple relationship satisfaction. The PFB-K consists of 10 Items, 9 of which measure on a 4-point Likert scale and the 10th item measuring on a 6-point Likert scale. All 9 items are averaged to a single value of overall satisfaction with the relationship. Reliability has been calculated to be between α = .85 and .91 [45]. The items cannot be shared due to copyright.

## Analysis

Our approach for data analysis was to establish a factor structure via confirmatory factor analysis (CFA), and assess reliability, measurement fit, and concurrent validity. The standard $p <$ .05 criterion was used to determine if factor loadings and correlations were significant. No further corrections were applied as all analyses contain different combinations of variables and a robust estimator was used. No transformations or item parceling was applied.

All data analyses were carried out in R Studio (version 4.3.3) [46], utilizing the following packages: *lavaan* (version 0.6–16) [47], *tidyverse* (version 2.0.0) [48], *eeptools* (version 1.2.5) [49], *lubridate* (version 1.9.2) [50], *psych* (version 2.3.6) [51], *naniar* (version 1.0.0) [52], *coefficientalpha* (version 0.7.2) [53], *Hmisc* (version 5.1–0) [54], *moments* (version 0.14.1) [55], and *e1071* (version 1.7–13) [56]. All packages were updated on the 20/08/2024. The study followed the recommendations of Jackson et al. [57] for the application of CFA. Little's MCAR test was used to inspect whether data was missing completely at random (MCAR). In our pre-

registration, we opted against imputation and for listwise deletion in the case of values MCAR, since we wanted to stick with genuine measurements.

A CFA was run with the original factor model of the PRFQ (Study 1) [3]. To assess model fit, the recommendations of Hu and Bentler [58] and Jackson et al. [57] were followed. We reported the $\chi^2$ value despite its tendency to be overly sensitive to sample size [59] but relied on the Comparative Fit Index (CFI), Standardized Root Mean Square Residual (SRMR), Root Mean Square Error of Approximation (RMSEA), and Tucker-Lewis Index (TLI) to determine model fit. The cut-off values for good model fit were CFI > .95, TLI > .95, SRMR < .08, and RMSEA < .06 [58]. A covariance matrix was chosen as input matrix and a robust maximum likelihood (MLR) estimator was used.

For the resulting model, reliability was assessed using McDonald's Omega ($\omega$) [60] and Cronbach's Alpha ($\alpha$) [61]. Values of > .70 were seen as indicative of a reliable scale [62].

The skewness, kurtosis and distribution were assessed. A parallel analysis using a 95th percentile criterion was run to inspect the suggested number of factors [63,64]. An exploratory factor analysis (EFA) with a maximum likelihood estimator, promax rotation, and the number of factors as suggested by parallel analysis was used to inspect item loadings. The subscale assignment for each item remained as in the original version, since we wanted to stick to the theoretically determined subscales.

Next, the sample was randomly split into two equal groups for training ($n$ = 189) and testing ($n$ = 189) a new model via CFA. Using the training data set, a total of 8 models were tested using lavaan's cfa() function [46]. The models incorporated varying factor correlations, residual item correlations, and deleted items (see Supplementary Table 9 in S4 Table for all tested models). We aimed to identify a model with at least an acceptable fit and having the least number of modifications. After having discovered a working model, it was cross-validated using the testing data set. Finally, the reliability was once again calculated using Cronbach's $\alpha$ and McDonald's $\omega$.

The concurrent validity was analyzed via Pearson correlations between the PRFQ, the ETMCQ, PSS, EPDS, age of child and parents, and relationship satisfaction, via a Spearman correlation between the PRFQ and education, and lastly via a point biserial correlation between the PRFQ and changes in the working environment. Effect sizes were presumed as small for $r$ > .10, medium for $r$ > .30, and $r$ > .50 [65]. An overview of the expected correlations can be seen in the online supplements. Additionally, we explored a bifactorial model. We employed a Schmid-Leiman transformation and an oblimin factor rotation, using the approach outlined in [66]. This model added a general factor to the original model. To assess model fit, the Root Mean Squared Residuals (RMSR) was employed, with the same cut-off as the standardized RMSR, < .09 for excellent model fit [58].

## Results

### Analysis of missing data, skewness, and kurtosis

The only missing data were the answers of 11 participants in the PFB-K, measuring relationship satisfaction. Little's MCAR test was non-significant ($\chi^2$ = 66.0, $df$ = 62, $p$ = .340), indicating that missing data was completely at random and list-wise deletion was applied. For the PM subscale, skewness was calculated at 2.14 and kurtosis at 3.86. For the CMS subscale, skewness was -0.20 and kurtosis -0.87. Lastly, for the IC subscale, skewness was -1.02 and kurtosis was 0.77.

### Confirmatory factor analysis of the original PRFQ model

Using the original model for a CFA, all items loaded significantly onto their respective factors, and the IC and CMS subscale correlated significantly ($r$ = .48, $p$ < .001). However, the model

showed poor fit [61]: CFI = .80, TLI = .78, SRMR = .09, RMSEA = .07 (CI [.06, .08]), $\chi^2$ = 374.62 ($p$ < .001). As such, the original model could not be replicated. The reliability assessment with Cronbach's $\alpha$ showed (1) $\alpha$ = .80 for CMS, (2) $\alpha$ = .66 for IC, and (3) $\alpha$ = .54 for PM. The reliability as analyzed by McDonald's $\omega$ returned (1) $\omega$ = .88 for CMS, (2) $\omega$ = .73 for IC, and (3) $\omega$ = .61 for PM. None of these values could be significantly improved by removing items. Further information on skew and distribution is available in the online supplement (for graphical representations see Supplementary Fig. 1–3 in S1 Fig). Given that the original model did not hold, all results beyond this point are directly based on the new model we propose for the German PRFQ.

## Model adaptation for the German translation

First, a parallel analysis was run to check the number of suggested factors. Both the parallel analysis and the scree plot suggested three factors, with Eigenvalues of 3.3 for factor 1, 1.3 for factor 2, and 0.8 for factor 3. Next, an EFA with the suggested three factors was used to inspect factor loadings of individual items (see Supplementary Table 10 in S5 Table). The CMS factor explained 16% of variance with loadings ranging from .43 to .74, the IC factor 8% with loadings between .13 and .71, and the PM factor 9% with loadings ranging from .19 to .65.

The sample was randomly split in two groups. The first group was used to find a new model by testing a total of 8 CFA models (see the online supplement for an overview of the models). Ultimately, a model that provided good fit was found (see Fig 1). After removing the second

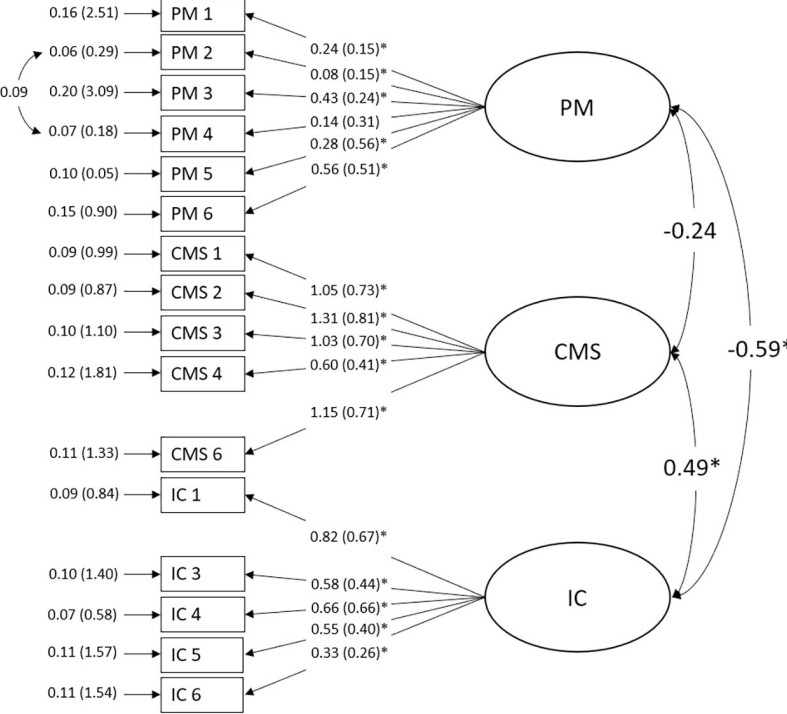

**Fig 1. Final factor model of the German PRFQ.** All items were retained regardless of factor loadings, given that we wanted to retain as much of the original content validity and theory base as possible. Only two items were removed to achieve model fit. Significant estimates are marked with*. Estimates in front of parathesis are unstandardized, estimates inside of parenthesis are standardized. Standard errors are in front of parentheses, error variances inside of parentheses. Unstandardized and standardized estimates for covariances were identical. Figure available at https://osf.io/j69wx/, under a CC-BY 4.0 license.

item of the IC subscale, the fifth item of the CMS subscale, adding an error covariance between the second and fourth item of the PM scale, and allowing the factors to freely correlate, all fit indices indicated an acceptable to good model fit: CFI = .94, TLI = .93, SRMR = .07, RMSEA = .04 (CI [.01, .06]). The $\chi^2$ value was 127.65 ($p$ = .032). The IC factor correlated significantly with the CMS factor ($r$ = .49, $p < .001$), as well as with the PM factor ($r$ = -.59, $p$ = .001). The correlation between CMS and PM was non-significant ($r$ = -.23, $p$ = .094). To cross validate the new 16-item model, the second half of the sample was then used to run another CFA. The fit indices showed acceptable to good model fit again: CFI = .92, TLI = .90, SRMR = .07, RMSEA = .05 (CI [.03, .06]). The $\chi^2$ value was 145.05 ($p$ = .002). The reliability values of the subscales, as measured by Cronbach's $\alpha$, pertaining to the new model, are as follows: 1. CMS: $\alpha$ = .79, 2. IC: $\alpha$ = .64, and 3. PM: $\alpha$ = .54. These values indicate a reliable CMS subscale. The reliability of the IC and PM subscale did not meet our reliability cutoff of $\alpha$ = .70. The reliability values of the subscales, as measured by McDonald's $\omega$, pertaining to the new model, are as follows: (1) CMS: $\omega$ = .84, (2) IC: $\omega$ = .69, and (3) PM: $\omega$ = .61. These values indicate a reliable CMS subscale. The reliability of both IC and PM fell short of our cut-off value of $\omega$ = .70.

### Alternative models

The bifactorial model using the same 16-item architecture was run with the addition of a general factor. The model yielded an RMSR of .25, indicating a misspecified model. Hence, this approach was not further pursued. Similarly, both two and four factor models yielded insufficient fit (CFI = .806, TLI = .774, SRMR = .085, RMSEA = .075 (CI [.060, .089] and CFI = .898, TLI = .879, SRMR = .069, RMSEA = .054 (CI [.039, .068]), respectively).

### Concurrent validity

Summarizing the assessment of the concurrent validity, 18 out of 30 preregistered correlations were correctly anticipated. All incorrectly anticipated correlations were in the small effect range (r = -.12 to .15). Only the correlation between CMS and perceived stress stood out, with an effect size of small to medium (r = -.23), indicating that individuals reporting higher levels of stress tended to struggle more with PRF. All correlations can be seen in Table 2.

## Discussion

The overall goals of the present study were (1.) to replicate the original PRFQ factor structure or (2.) to propose a new model for the German PRFQ via exploratory means, as well as (3.) to assess the reliability of the German PRFQ and (4.) to analyze its concurrent validity.

### Interpretation and comparisons of the results

**Model replication and alternative proposition.** The original factor model of Luyten et al. [3] could not be replicated and the CFI, TLI, and SRMR indicated that the model was misspecified [58]. Notably, given that Luyten et al. [3] had to add a number of not specified error covariances to achieve model fit, we were not able to test the exact same model. The remaining discussion will only address our final model for the German PRFQ.

**Alternative proposition of a new model.** Regarding a new model for the German PRFQ, parallel analyses suggested a three-factor model. A subsequent EFA with three factors was run, following the recommendations by Hair et al. [67] and considering factor loadings of .30 to be significant. For CMS, all items loaded significantly onto the subscale, along with item 5 cross-loading onto the PM subscale. For IC, items 1–5 loaded onto IC, item 4 additionally cross-loaded onto PM and item 6 did not significantly load onto any of the subscales. A number of

**Table 2. Correlational analyses assessing concurrent validity.**

| Questionnaire | CMS | IC | PM |
|---|---|---|---|
| Trust (ETMCQ) | N.s. | N.s. | Neg. Cor. |
| Pearson Cor. | .01 | .03 | .12 |
| | (.861) | (.555) | (.020) |
| Mistrust (ETMCQ) | Neg. Cor. | Negligible | Pos. Cor. |
| Pearson Cor. | -.01 | .10 | .18 |
| | (.851) | (.058) | (< .001) |
| Credulity (ETMCQ) | Neg. Cor. | Negligible | Pos. Cor. |
| Pearson Cor. | .07 | .07 | .11 |
| | (.163) | (.188) | (.028) |
| Perceived Stress (PSS) | Negligible | N.s. | Pos. Cor. |
| Pearson Cor. | -.23 | -.12 | .27 |
| | (< .001) | (.022) | (< .001) |
| Postpartum Depression (EPDS) | Negligible | N.s. | Pos. Cor. |
| Pearson Cor. | -.07 | -.04 | .26 |
| | (.161) | (.454 | (< .001) |
| Age of Parents | N.s. | N.s. | Negligible |
| Pearson Cor. | -.06 | -.14 | .05 |
| | (.285) | (.006) | (.362) |
| Age of Child | N.s. | N.s. | N.s. |
| Pearson Cor. | .00 | -.10 | .15 |
| | (.924) | (.047) | (.003) |
| Level of Education | N.s. | N.s. | Neg. Cor. |
| Spearman Cor. | -.12 | .07 | -.10 |
| | (.024) | (.195) | (.042) |
| Relationship Satisfaction (PFB-K) | Pos. Cor. | Pos. Cor. | Neg. Cor. |
| Pearson Cor. | .10 | .13 | -.09 |
| | (.045) | (.009) | (.083) |
| Changes in the Working Environment | N.s. | N.s. | Pos. Cor. |
| Point Biserial Cor. | -.02 | .01 | -.01 |
| | (.736) | (.901) | (.886) |

The first line of each cell contains outcomes as predicted by the pre-registration (https://osf.io/j69wx/), with options being a predicted positive correlation (Pos. Cor.), negative correlation (Neg. Cor.), negligible meaning significant but $r \leq .10$, and finally not significant (N.s.). The second line contains the correlation type and coefficient, and lastly the third line contains the $p$-value. Italic correlations were in line with the predictions. Correlations which were assumed to be negligible but ended up being non-significant were still deemed correctly predicted.

other translations also had issues with factor loadings with item 6 [20,27,28]. Lastly, for PM, all items but number 3 loaded on the subscale. The EFA showed all item loadings as intended, with cross-loadings being rare and if present only low in effect size. It was therefore concluded that the EFA did not warrant modifications to the factor structure.

The new model demonstrated good fit as judged using RMSEA and SRMR and acceptable fit using TLI and CFI (see 1). This model contained the same three factors as the original model but allowed free correlation. The second item of the IC scale ("I wonder a lot about what my child is thinking and feeling.") and the fifth item of the CMS scale ("I always know why I do what I do to my child.") were removed. Lastly, we had to add one error covariance between the second and fourth PM items. Ideally, the latter measure would have been avoided, but since the items came from the same subscale and correlated, this was deemed acceptable.

The $\chi^2$ value was significant, indicating misspecification. However, given this metric's tendencies to dismiss adequate models in larger sample sizes [59], coupled with the results from the other four fit indices, we assumed our model to show good to excellent fit. In our cross-validation, RMSEA and SRMR again indicated good fit and TLI and CFI indicated acceptable fit, which suggests no overfitting. This cross-validation together with our conservative approach with cut-off values further strengthens our final model and supports the original factor model [3].

Interestingly, the Finnish validation [23] also excluded item 2 from the IC subscale. A possible explanation might be its double-barreled wording ("I wonder a lot about what my child is thinking and feeling."), asking about both the cognitive component (thinking) and the affective component (feeling). Additionally, the German translation "Ich frage mich häufig. . ." might come across as more serious and obsessive as compared to the original English "I wonder a lot. . .". The fifth CMS item is the only item concerned with the mental state of the parent, not the child ("I always know why I do what I do to my child."). While theoretically still part of PRF, it is a clear difference from the other items. The Certainty About Mental States Questionnaire (CAMSQ), a self-report measure designed to only assess CMS, also differentiates between self- and other-certainty, and achieves good reliability [68]. The CAMSQ therefore gives empirical evidence that the two abilities of mentalizing about oneself and others are related, but not the same, and subsequently why the removal of this item improves fit. However, while the CAMSQ [68] only captures the dimension of certainty about mental states, which is also captured by the CMS subscale in the PRFQ, the PRFQ provides the possibility to capture two additional dimensions of PRF. Another issue could be the wording of the item "I always know why I do what I do to my child." in the German translation ("Ich weiß immer genau, warum ich das mit meinem Kind tue, was ich tue"), which changes the item from "i always know why. . ." to "i always know *exactly* why. . .". Thus, we advise that researchers should reassess this item after having removed the "genau" (= exactly) and then rerun the model to check if this change makes the item viable again. Looking back to the EFA results, this item was also the only one cross-loading in that particular subscale.

While the prospect of having a bifactorial model would have opened some exciting avenues for future research, the RMSR value that we obtained from our analysis could not support this modeling approach.

**Reliability analysis.** Judging by Cronbach's α and McDonald's ω, the CMS subscale showed good reliability, while the reliabilities of the IC and PM subscale needs further assessment and improvement. These findings are still mostly on par with other validations of the PRFQ. While PM only barely made the cut-off of .70 in the original validation (α = .70), it was below the cut-off in the Italian, Korean, and Finnish validations [21,23,24], with the Italian version also struggling with the reliability of IC. In its first application [26], the 18-item version of the German PRFQ struggled with reliability of the IC (α = .47) and PM (α = .52) subscale. On the other hand, the Portuguese [22] and Canadian validations [20] showed reliable measures of all three subscales.

Considering these results as well as those of prior studies, the internal consistencies of the IC and PM subscales appear to be an issue in most languages [69]. A possible explanation for this might be found in the complexity of the construct, reflected in its measurement using complex coding of interviews and multi-factored self-report questionnaires. PRF and Mentalizing require both affective and cognitive processes [70], meaning that parts of these particular mental abilities rely on unconscious processes [16] which might not be easily captured by self-report, especially PM. Another consideration is social desirability, particularly with some very strongly worded PM items. This might partly explain the extreme skew of the PM subscale. However, this subscale has in part been developed for clinical risk samples. Given that we

investigated a general, non-clinical sample, low average scores may be expected. Overall, the German PRFQ still shows acceptable performance in terms of reliability.

*Concurrent Validity Analysis.* The majority of our predictions about concurrent validity were in line with our results (18 out of 30 hypotheses tested). The negative correlation between CMS and perceived stress stood out. However, both the Canadian validation [20] and Luyten et al. [3] also showed that individuals reporting higher levels of stress also scored lower on PRF, however their effects remained insignificant. One possible reason for this might be the increase in stress during the COVID-19 pandemic, since high stress might impact mentalizing capacity [71,72]. Perceived stress increased by more than a standard deviation in our sample [73] compared to German mothers pre-pandemic (M = 22.29, SD = 3.43 vs. M = 13.07, SD = 6.08) [74]. Therefore, contextual effects on PRF cannot be ruled out. Another source for the deviation could be that, following the theory behind the CMS scale, it should be mapped non-linearly, since both exceedingly high and low values should be considered unfavorable (hyper- and hypomentalizing respectively). This mismatch between theory and practice might also cause artifacts in our correlation, though it would not explain why this has not been an issue for other validation studies of the PRFQ. Finally, taking all analyses into consideration, it can be assumed that the German PRFQ demonstrated adequate concurrent validity.

## Limitations

The main limitations of the present study stem from its sample, mainly its high SES background. The majority of parents were in the two highest income brackets and also had the highest achievable school degree. For further validation, it would be desirable to also include lower SES backgrounds. Additionally, the data we used did not contain any information on ethnic identity. Second, the sample was highly motivated, since this was the third time they participated in the longitudinal study, meaning some self-selection has to be assumed. As a third limitation, the reliability of the presented questionnaire still poses an issue and requires further clarification.

## Recommendations for further research

To increase reproducibility [75], we encourage the use of open science methods. The majority of other validations provided no preregistration or open data, which made replication significantly harder. By providing these additional materials, subsequent studies have a stronger foundation to build upon.

Another issue is the scaling of the CMS subscale. The IC and PM subscales are mostly modeled in a linear fashion. However, as already discussed, CMS might follow an inverted U-shape function because of hyper- and hypomentalizing being situated on the opposing ends. A first piece of evidence for this approach might be found in the fundamental difference in distribution between the subscales. In our study, the IC subscale skews towards high values and the PM subscale towards low values. But for CMS, the majority of parents indicated middle values. This suggests that there might be a fundamental difference in how the scaling of the three subscales works. As such, a more appropriate way for future research to analyze CMS might be an unfolding model.

## Conclusion

We propose a 3-factor, 16-item model for the German adaptation of the PRFQ. We managed to show acceptable to good fit, concurrent validity, and good reliability for the CMS subscale, while the reliability of the IC and PM subscale needs further assessment and improvement. The 16-item German PRFQ has been proven a valid and promising self-report measure for

future studies into attachment and psychopathology. Given the complex nature of mentalizing and PRF, we encourage further validation studies of our German version of the PRFQ by other independent research groups.

## Supporting information

**S1 File. Factor structure differences between PRFQ translations.**
(DOCX)

**S1 Fig. Histograms of the German PRFQ Subscales.**
(DOCX)

**S1 Table. Predicted correlations and hypotheses for concurrent validity.**
(DOCX)

**S2 Table. Item translations of all used questionnaires.**
(DOCX)

**S3 Table. Sample means and standard deviations for all questionnaires.**
(DOCX)

**S4 Table. Overview of all tested models and their respective fit indices.**
(DOCX)

**S5 Table. Factor loadings as obtained by the EFA.**
(DOCX)

## Acknowledgments

We would like to thank all volunteers who participated in the online survey and all colleagues who contributed to this study and article.

## Author Contributions

**Conceptualization:** Andreas S. Wildner, Tobias Nolte, Corinna Reck, Peter Fonagy, Patrick Luyten, Anna-Lena Zietlow, Christian F. J. Woll-Weber.

**Data curation:** Andreas S. Wildner, Su Mevsim Küçükakyüz, Anton K. G. Marx.

**Formal analysis:** Andreas S. Wildner, Su Mevsim Küçükakyüz, Christian F. J. Woll-Weber.

**Investigation:** Andreas S. Wildner, Su Mevsim Küçükakyüz, Anton K. G. Marx, Alexandra von Tettenborn, Christian F. J. Woll-Weber.

**Methodology:** Andreas S. Wildner, Su Mevsim Küçükakyüz, Anton K. G. Marx, Mitho Müller, Christian F. J. Woll-Weber.

**Supervision:** Christian F. J. Woll-Weber.

**Visualization:** Andreas S. Wildner, Su Mevsim Küçükakyüz, Christian F. J. Woll-Weber.

**Writing – original draft:** Andreas S. Wildner, Su Mevsim Küçükakyüz, Christian F. J. Woll-Weber.

**Writing – review & editing:** Anton K. G. Marx, Tobias Nolte, Corinna Reck, Peter Fonagy, Patrick Luyten, Alexandra von Tettenborn, Mitho Müller, Anna-Lena Zietlow.

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
