## [Decision Letter · Decision Letter 0]

12 Jun 2024

PONE-D-24-14873Parental Considerations About Their Childs’ Mental Health: Validating the German Adaptation of the Parental Reflective Functioning QuestionnairePLOS ONE

Dear Dr. Wildner,

Thank you for submitting your manuscript to PLOS ONE. After careful consideration, we feel that it has merit but does not fully meet PLOS ONE’s publication criteria as it currently stands. Therefore, we invite you to submit a revised version of the manuscript that addresses the points raised during the review process.

We look forward to receiving your revised manuscript.

Kind regards,

Ted C.T. Fong, Ph.D.

Academic Editor

PLOS ONE

Additional Editor Comments:

Specific comments to Author:

- Line 164: Remove the model fit criteria "e CFI > .90, TLI > .90, SRMR < .09, and RMSEA < .08", the criteria on line 165 "CFI > .95, TLI > .95, SRMR < .08, and RMSEA < .06" refers to good model fit. Same for the model fit on lines 186-187.

- Report the skewness and kurtosis of the PRFQ items. Were they treated as continuous items in the factor analysis? Please state clearly.

- The authors should present the final factor model apart from the proposed factor model so that the readers could see the changes more clearly.

- In Figure 1, quite a number of items showed low factor loadings (< 0.5). Were they retained in the final model or not? And why?

- Model comparison should be done to compare the model fit of the 3-factor model to alternative 2-factor and 4-factor models.

Reviewers' comments:

Reviewer's Responses to Questions

**Comments to the Author**

1. Is the manuscript technically sound, and do the data support the conclusions?

Reviewer #1: Partly

Reviewer #2: Yes

Reviewer #3: Partly

2. Has the statistical analysis been performed appropriately and rigorously? 

Reviewer #1: No

Reviewer #2: Yes

Reviewer #3: N/A

3. Have the authors made all data underlying the findings in their manuscript fully available?

Reviewer #1: Yes

Reviewer #2: Yes

Reviewer #3: Yes

4. Is the manuscript presented in an intelligible fashion and written in standard English?

Reviewer #1: No

Reviewer #2: Yes

Reviewer #3: Yes

5. Review Comments to the Author

Reviewer #1: ID: PONE-D-24-14873

Title: Parental Considerations About Their Childs’Mental Health: Validating the German Adaptation of the Parental Reflective Functioning Questionnaire

Thank you for providing a chance to review this manuscript.

Detailed information:

Abstract

1) Please state the purpose of your research.

2) Abstracts require a brief summary of the article's background, aims, methods, results and conclusions. The current abstract does not provide a good overview, especially of the results. The results are meant to be narrated with statistically relevant data. It is suggested that subheadings could be added to the abstract to make the whole abstract clearer and easier to understand. Rewrite it with reference to high quality literature.

Method

Participants & Procedure

Line 116, Page 7: “All data used in this study (34) stems from the third measuring time point” What is the reason for choosing time point 3 instead of 1,2?

Line 112-127, Page 6-7: Please describe the underlying information such as the location of the survey.

Line 120, Page 7: 1) “The target population were mothers of children aged 0 to 3 years” Is this all the inclusion criteria? Are there exclusion criteria for the population, if so, please add them. 2) “which were recruited for the study by various social media channels, medical institutions and professionals, and professional support organizations.” It is recommended to list how many people have been recruited by each route, as there is still a big difference between the two methods of recruitment.

Line 121, Page 7: “ by various social media channels” Is it possible to give some examples of social media?

Line 123, Page 7: “missing data” What are the missing values? What is the reason why you don't use the missing value handling method but just delete it?

Line 122-127, Page 7: Has the sample size of 378 been done enough? How was the sample size calculated? What is the sample missingness rate? What was the sampling method for the population? Please specify the process of recruiting the sample.

Line 125, Page 7: “yrs” For the first occurrence in the text, do not use abbreviations; use the full name.

Line 126, Page 7: “(SD = 13.32, range = 10.2 - 78.6).” The previous article stated that mothers of children aged 0-3 were surveyed, is this age range too out of line?

Materials

Line 128, Page 7: One subtitle for each tool (Measure).

Line 135-136, Page 7: “Next to the PRFQ, the following variables and corresponding instruments were chosen to assess concurrent validity” All scales are part of the analysis and therefore all need to be described in detail, including basic information about the scale and the reliability of the translated version.

Line 145, Page 8: 1) It is recommended to align the numbers in the table so that it is more aesthetically pleasing. 2) Currencies are to be explained and should be written out in Notes like an acronym.

Analysis

Line 149-151, Page 9: This paragraph is not appropriate for this section.

Line 153-154, Page 9:1) all packages italicized with version number. 2) The exact date of acquisition should also be stated

Line 157-167, Page 9-10: “Next” The description can be done in segments here and the narrative is in the wrong order, it is a confidence test under a CFA determined model, please replace the order.

Line 171-172, Page 10: How much sample size for each of the two groups is something that needs to be stated in the article.

Line 188-189, Page 11: This paragraph would be more appropriately placed at the beginning.

There are some problems in this paper: 1) much of the content is not detailed enough; 2) there are some logical errors.

Thank you and my best,

Your reviewer

Reviewer #2: The paper assesses the structural and concurrent validity of the German version of PRFQ. It is well written. Few comments for the authors.

Title

It may be advisable to let the type of population included be reflected in the title. i.e. healthy children or not.

Keywords: Can you add validation to keywords?

Introduction

Write in full PRF in the sub-title "PRF and its Relevance".

Methods

i. Please expand on the inclusion and exclusion criteria.

ii. Apart from age, it will be better to give clinical characteristics of these children. Are they healthy children?

Other comments as indicated in the attached.

Reviewer #3: The present study investigates the Parental Reflective Functioning Questionnaire (PRFQ) and its adaptation for the German population. While the paper is interesting, it faces several issues regarding its statistical analysis. Here are my main comments:

- It is unclear how the authors accounted for the fact that the items are on a Likert scale, and therefore are not quantitative variables, in their Exploratory Factor Analysis (EFA) and Confirmatory Factor Analysis (CFA) estimations. For example, the maximum likelihood estimation used in EFA typically assumes normality.

- How did the authors assess the skewness (as they mentioned) in ordinal variables?

- What correlation coefficient is presented in Table 2? How were the subscales determined? Was item parceling or factor scores used? It is important to note the issues that can arise with item parceling.

- Regarding the assessment of the instrument's concurrent validity: do the other instruments used in the study have good psychometric properties?

- What are the estimates shown in Figure 1? Where are the error variances? The unstandardized estimates, standard errors, standardized estimates for each path, and the variance of the error variables must be included.

- What criteria were used to exclude missing cases? Was any other data screening procedure adopted, such as assessing for careless responding?

- How was the translation of items into German validated?

6. PLOS authors have the option to publish the peer review history of their article (what does this mean?). If published, this will include your full peer review and any attached files.

Reviewer #1: No

Reviewer #2: No

Reviewer #3: No

---

## [Author Response · Author response to Decision Letter 0]

24 Aug 2024

Dear editors and reviewers,

many thanks for all the attentive and constructive comments! The manuscript has been changed to comply with the journal requirements. All content related comments have been addressed in the rebuttal letter. We look forward to hearing from you!

Best,

Andreas Wildner

---

## [Decision Letter · Decision Letter 1]

22 Sep 2024

PONE-D-24-14873R1Parental considerations about their childs’ mental health: validating the german adaptation of the Parental Reflective Functioning QuestionnairePLOS ONE

Dear Dr. Wildner,

Thank you for submitting your manuscript to PLOS ONE. After careful consideration, we feel that it has merit but does not fully meet PLOS ONE’s publication criteria as it currently stands. Therefore, we invite you to submit a revised version of the manuscript that addresses the points raised during the review process.

Your revised work has been reviewed by the three original reviewers, who are mostly satisfied with the revision. Please amend the manuscript according to the remaining reviewer comments before we can make a final decision on acceptance.

We look forward to receiving your revised manuscript.

Kind regards,

Ted C.T. Fong, Ph.D.

Academic Editor

PLOS ONE

Journal Requirements:

Reviewers' comments:

Reviewer's Responses to Questions

**Comments to the Author**

1. If the authors have adequately addressed your comments raised in a previous round of review and you feel that this manuscript is now acceptable for publication, you may indicate that here to bypass the “Comments to the Author” section, enter your conflict of interest statement in the “Confidential to Editor” section, and submit your "Accept" recommendation.

Reviewer #1: (No Response)

Reviewer #2: All comments have been addressed

Reviewer #3: All comments have been addressed

2. Is the manuscript technically sound, and do the data support the conclusions?

Reviewer #1: Partly

Reviewer #2: (No Response)

Reviewer #3: Yes

3. Has the statistical analysis been performed appropriately and rigorously? 

Reviewer #1: Yes

Reviewer #2: (No Response)

Reviewer #3: Yes

4. Have the authors made all data underlying the findings in their manuscript fully available?

Reviewer #1: Yes

Reviewer #2: (No Response)

Reviewer #3: Yes

5. Is the manuscript presented in an intelligible fashion and written in standard English?

Reviewer #1: Yes

Reviewer #2: (No Response)

Reviewer #3: Yes

6. Review Comments to the Author

Reviewer #1: ID: PONE-D-24-14873R1

Title: Parental considerations about their childs’ mental health: validating the german

adaptation of the Parental Reflective Functioning Questionnaire

Thank you for providing a chance to review this manuscript.

Detailed information:

Participants & Procedure

1) “which were recruited for the study by various means such as social media channels (i.e., X/Twitter, Facebook and Instagram posts), medical institutions and professionals, and professional support organizations, with social media being the most useful tool.” Please indicate the number of persons recruited by each route.

The article has been very much revised, though there are still a handful of problems, and the author is asked to change it carefully.

Thank you and my best,

Your reviewer

Reviewer #2: (No Response)

Reviewer #3: (No Response)

7. PLOS authors have the option to publish the peer review history of their article (what does this mean?). If published, this will include your full peer review and any attached files.

Reviewer #1: No

Reviewer #2: No

Reviewer #3: No

---

## [Author Response · Author response to Decision Letter 1]

25 Oct 2024

Comment: 

Participants & Procedure

1) “which were recruited for the study by various means such as social media channels (i.e., X/Twitter, Facebook and Instagram posts), medical institutions and professionals, and professional support organizations, with social media being the most useful tool.” Please indicate the number of persons recruited by each route.

Answer:

Thank you for your remark! In our previous rebuttal letter, we explained that, unfortunately, we were unable to include a question about how participants heard about the study due to space constraints in the questionnaire. While we recognize the importance of recruitment details, we had to omit this and other items to avoid overburdening participants. As a result, we cannot precisely report how participants were recruited. However, we were able to pro-vide a rough estimate of the distribution based on when participants reached out (lines 144-146). We chose to report percentages rather than total numbers to avoid creating a false sense of accuracy.

---

## [Decision Letter · Decision Letter 2]

5 Nov 2024

Parental considerations about their childs’ mental health: validating the german adaptation of the Parental Reflective Functioning Questionnaire

PONE-D-24-14873R2

Dear Dr. Wildner,

We’re pleased to inform you that your manuscript has been judged scientifically suitable for publication and will be formally accepted for publication once it meets all outstanding technical requirements.

Kind regards,

Ted C.T. Fong, Ph.D.

Academic Editor

PLOS ONE

Additional Editor Comments (optional):

Reviewers' comments:

Reviewer's Responses to Questions

**Comments to the Author**

1. If the authors have adequately addressed your comments raised in a previous round of review and you feel that this manuscript is now acceptable for publication, you may indicate that here to bypass the “Comments to the Author” section, enter your conflict of interest statement in the “Confidential to Editor” section, and submit your "Accept" recommendation.

Reviewer #1: All comments have been addressed

2. Is the manuscript technically sound, and do the data support the conclusions?

Reviewer #1: Yes

3. Has the statistical analysis been performed appropriately and rigorously? 

Reviewer #1: Yes

4. Have the authors made all data underlying the findings in their manuscript fully available?

Reviewer #1: Yes

5. Is the manuscript presented in an intelligible fashion and written in standard English?

Reviewer #1: Yes

6. Review Comments to the Author

Reviewer #1: ID: PONE-D-24-14873R2

Title: Parental considerations about their childs’ mental health: validating the german

adaptation of the Parental Reflective Functioning Questionnaire

Thank you for providing a chance to review this manuscript.

Recommendation: Accept.

After careful revision by the authors, the manuscript has met the criteria for publication, congratulations!

Thank you and my best,

Your reviewer

7. PLOS authors have the option to publish the peer review history of their article (what does this mean?). If published, this will include your full peer review and any attached files.

Reviewer #1: No

---

## [Editor Report · Acceptance letter]

22 Nov 2024

PONE-D-24-14873R2 

PLOS ONE

Dear Dr. Wildner, 

I'm pleased to inform you that your manuscript has been deemed suitable for publication in PLOS ONE. Congratulations! Your manuscript is now being handed over to our production team.

Kind regards, 

on behalf of

Dr. Ted C.T. Fong 

Academic Editor

PLOS ONE